# Immunopathology of Extracellular Vesicles in Macrophage and Glioma Cross-Talk

**DOI:** 10.3390/jcm12103430

**Published:** 2023-05-12

**Authors:** Timothy J. Kopper, Xiaoli Yu, Michael W. Graner

**Affiliations:** Department of Neurosurgery, University of Colorado Anschutz Medical Campus, 12700 E 19th Ave., Aurora, CO 80045, USA; timothy.kopper@cuanschutz.edu (T.J.K.); xiaoli.yu@cuanschutz.edu (X.Y.)

**Keywords:** glioblastoma, macrophage, microglia, extracellular vesicles, exosomes

## Abstract

Glioblastomas (GBM) are a devastating disease with extremely poor clinical outcomes. Resident (microglia) and infiltrating macrophages are a substantial component of the tumor environment. In GBM and other cancers, tumor-derived extracellular vesicles (EVs) suppress macrophage inflammatory responses, impairing their ability to identify and phagocytose cancerous tissues. Furthermore, these macrophages then begin to produce EVs that support tumor growth and migration. This cross-talk between macrophages/microglia and gliomas is a significant contributor to GBM pathophysiology. Here, we review the mechanisms through which GBM-derived EVs impair macrophage function, how subsequent macrophage-derived EVs support tumor growth, and the current therapeutic approaches to target GBM/macrophage EV crosstalk.

## 1. Introduction

Glioblastomas (CNS5 WHO Grade 4 astrocytomas, IDH-wildtype, henceforth called GBM) are a devastating disease, with extremely poor clinical outcomes evidenced by the median survival time of 15 months post-diagnosis [1]. New therapies are desperately needed; however, many recent approaches have failed to improve outcomes [2,3,4,5], including immunotherapeutic approaches. Insufficient understanding of the tumor microenvironment is a critical challenge impairing the development of effective therapies. Resident (microglia) and infiltrating macrophages are a substantial component of the tumor environment and the tumor itself, constituting 30–50% of the cellular content [6,7], and are primary mediators of inflammatory responses [8]. Macrophages are a key cellular component of a successful anti-tumor response in which they identify and phagocytose cancerous tissues [9,10]. In GBM and other cancers, this capability is impaired or blocked entirely by tumor-induced suppression of anti-tumor immune responses [11,12,13]. In an increasing number of instances, this is recognized as being induced through extracellular vesicle (EV) intercellular communication [12]. EVs are lipid-enclosed vesicles released from cells that contain proteins, nucleic acids, and other biological mediators, allowing for intercellular communication in both health and disease [14,15]. GBM/macrophage EV crosstalk is a critical component of the tumor microenvironment [11,12]. With improved understanding, these interactions could be harnessed—or discouraged—to engage endogenous anti-tumor responses and develop novel targeted therapies. Here, we review the current state of GBM/macrophage EV crosstalk in the immunopathology of GBM.

## 2. Macrophage/Microglial Polarization in GBM Pathology

Macrophages are a key component of the innate and adaptive immune response. Circulating monocytes exit the bloodstream into numerous tissues and mature into resident macrophage populations throughout the body [16]. Microglia, the resident macrophage population in the central nervous system, are unique in that this population is established during embryonic development [17]. Functionally, however, monocyte-derived macrophages and microglia retain extensive similarities. In innate immunity, macrophages contain numerous receptors and sensors to identify the presence of infection, cellular damage, or cancer [16]. These macrophages can then induce numerous signaling cascades, inducing localized inflammation (production of reactive oxygen species, lymphocyte and Natural Killer (NK) cell activation, recruitment of neutrophils from the bloodstream, etc.). Similarly, these macrophages can induce systemic responses, such as a fever [16]. Once the activating stimulus is resolved, macrophages also produce factors contributing to cellular repair and the resolution of inflammation [18]. In adaptive immunity, macrophages are an important antigen-presenting cell which identify microbial antigens, or in the case of cancer, antigens from tumors that are either not expressed by healthy cells or antigens that are expressed much more highly in tumor cells [16]. Through this antigen expression, macrophages are able to activate adaptive immune responses, notably T-cells, to specifically target that antigen throughout the body [19]. Unfortunately, in tumors, the cancer often adapts to evade and manipulate the immune response. For example, tumors can begin to produce proteases, clipping off the ligand necessary for the macrophage-activated NK cells to attack the tumor cells [20]. Further, the tumor can signal directly to nearby macrophages to manipulate their activities, as discussed in detail in this review.

Macrophages are an extremely malleable cell type capable of adopting a wide spectrum of activation states [21,22]. These activation states control many aspects of macrophage physiology, including morphology, phagocytic activity, cytokine production, inflammatory profiles, and many other components of the immune response [23,24]. While macrophage activation is complex and multifaceted, for practical purposes this is often simplified to a pro-inflammatory “M1” subset and a second anti-inflammatory “M2” subset. The M1/M2 nomenclature is overly simplistic and does not cleanly apply across most disease states [25]; however, for general scientific discourse, it remains the best option available so long as its limitations are understood. In GBM, it is generally understood that pro-inflammatory “M1”-polarized macrophages promote a robust immune response and anti-tumor effects, whereas anti-inflammatory “M2”-polarized macrophages promote pro-tumor effects and dampen the broader immune response [13,26]. M0 is sometimes used in reference to naïve unstimulated macrophages [27]. Intracellularly, these macrophage activation states are regulated by several canonical and non-canonical signaling pathways. For M1 macrophages, this is typically via the NF-κβ/STAT1 [28,29] signaling axes, often modeled in vitro using lipopolysaccharide and interferon gamma stimulation of TLR-4 and the interferon-gamma receptor [30]. Conversely, M2 macrophages are more typically associated with the STAT3/STAT6 signaling axes [28], modeled with Interleukin 4 and/or 13 stimulation [31]. In the context of GBM, however, countless potential environmental stimuli likely induce complex activation states not fully in line with the M1/M2 nomenclature [32,33]. Further, given that macrophages are highly impressionable by extremely localized environmental factors, there are likely diverse activation states across the GBM macrophage populations [21]. Continued advances in single-cell RNA sequencing technologies will allow for an improved understanding of how the macrophage activation spectrum impacts GBM pathophysiology [34].

Critically, in the context of GBM, macrophage activation status determines whether the macrophage attacks the cancerous tissue or promotes continued growth and broader immune suppression [17,25]. M1 macrophages express high levels of antigen-presenting MHC complexes, through which they can display tumor-derived antigens and induce robust adaptive immune responses against the tumor [35,36]. M1 macrophages also express elevated levels of the costimulatory molecule CD87 (B7-2) allowing for improved activation of naïve T-cells [31]. Similarly, M1 macrophages produce elevated levels of cytotoxic factors such as reactive oxygen species, nitric oxide, and pro-inflammatory cytokines to directly target nearby tumor cells. Conversely, M2 macrophages are generally anti-inflammatory and immunosuppressive [26]. With reduced MHC and costimulatory molecule expression, M2 macrophages have a reduced ability to activate the adaptive immune response [35]. Further, under certain conditions, M2 macrophages can directly promote tumor growth in response to tumor signals, as discussed in this review. GBM macrophages are a heterogenous cell population derived from both tissue-resident microglia and infiltrating myeloid-derived macrophages. While these populations share many similarities and are often referred to collectively as macrophages, some distinctions can occur [25,37]. When examined separately experimentally, specific terminology will be utilized for clarity.

## 3. EVs as Biological Mediators of Intercellular Communication

EVs are small lipid-enclosed vesicles released from cells into the extracellular space [38,39]. EVs contain a broad mixture of biological materials, including proteins, lipids, nucleic acids, and other metabolites, whose contents can vary depending on the cellular source and EV subtype [40]. Even the lipid composition of the EV membrane can influence biological activity [41]. Historically, EVs were further distinguished into several EV subtypes based on size and cellular source or biogenesis. Exosomes are a subtype of EVs derived from the endosomal/multivesicular body system and range from 30–150 nm in diameter. Microvesicles are a subtype of EVs formed from the plasma membrane through outward budding and range from 100 nm to 1 µm in diameter. Apoptotic bodies form from dying cells. Notably, apoptotic bodies differ in that they contain additional cellular material, such as intact organelles and chromatin [42]. In recent years, the EV research field has updated its guidelines on EV terminology [43]. When specified in the source literature, the original terminology will be used in this review; however, where sufficient method detail is provided, we will also include the current terminology. Notably, these recent guidelines describe EVs in terms of their physical characteristics, molecular composition, and cellular origin via biogenesis [43]. Here, this will support a shift in terminology from the previous term “exosome” towards the modern terminology, “small EVs” (sEVs) for EVs under 200 nm diameter. We note that the biogenesis of “exosomes” versus “microvesicles” is seldom known when collecting EVs from biofluids or conditioned cell/tissue culture media, and thus the term “EVs” refers to both such populations [44].

While EVs were originally thought to be a form of cellular waste, it has since been appreciated that EVs conduct an intricate form of cellular communication [40]. They allow for cellular crosstalk both locally and across great distances [45]. These communications occur both in normal physiological conditions and in disease [15]. GBM tumor cells show an increased production of EVs relative to healthy tissue [46,47]. Critically, in GBM, EVs are thought to be a primary means through which the tumor manipulates its surroundings to invoke a more conducive environment for further growth and anti-cancer drug resistance [44,48,49]. Here, we review the mechanisms of GBM/macrophage EV communications and the implications of these signals in GBM immunopathology. Table 1 provides a brief summary of the key mechanisms through which GBM tissue and macrophages exchange EV crosstalk.

## 4. Impact of GBM-Derived EVs on Macrophage Function

EVs are a critical mechanism utilized by GBM tissue to evade the immune system [59]. Macrophages are the predominant immune cells in the tumor vicinity; however, they are ineffective at targeting tumors [60]. Worse, some macrophages begin to support the tumor, allowing for increased and faster growth [61]. GBM tissue supports this maladaptive macrophage response through numerous EV-dependent mechanisms as detailed below.

In 2015, de Vrij et al. applied GBM-derived EVs (presumably small EVs) onto monocyte-derived macrophages and observed several changes [48] (Table 1). Notably, the application of GBM-derived EVs resulted in an increased expression of CD163, a marker associated with the M2 pro-tumor phenotype, as measured by flow cytometry. The macrophages were found to produce increased levels of VEGF and IL-6. These cytokines have been previously found to support tumor growth (IL-6) [62] and tumor hyper-vascularization (VEGF) [63]. The authors hypothesized that these effects were caused by small RNA molecules enriched within GBM EVs, as had been demonstrated elsewhere with EVs from different tumor types [64]; however, they did not identify an exact candidate. Work by van der Vos et al. (2016) supported this hypothesis by visualizing the extensive uptake of GBM EVs (presumably small EVs) by primary microglia and noting the increase in GBM EV-cargo microRNAs in the microglia and macrophage. These studies suggest the transfer of microRNAs from EVs to the microglia/macrophage [65].

Zhao et al. (2022) also demonstrated an M2 polarization shift in macrophages treated with GBM EVs (presumably small EVs). They found that GBM EVs are enriched in microRNA-27a-3p and have even higher levels under hypoxic conditions (Figure 1, Table 1). Further, they demonstrated that microRNA-27a-3p inhibits enhancer of zeste homologue 1 (*EZH1*) [50]. Inhibition of *EZH1* has been previously demonstrated to promote M2 macrophage polarization [66]. Indeed, the authors observed that microRNA-27a-3p-treated macrophages had elevated levels of Arginase-1 (M2 phenotypic marker) and reduced iNOS (M1 phenotypic marker). Myeloid-cell-derived arginase-1 (via arginine depletion) has a noted ability to suppress T-cell responses [67]. A similar study confirmed the transfer of miR-21 from GBM to microglia in an in vivo mouse model and observed substantial microglial reprogramming [51]. This was an important study in that the host mice were miR-21 null, directly implying the microRNA was transferred from GBM EVs to cells of the tumor microenvironment. This is further evidence that GBM EVs may be preferentially taken up by tumor-associated macrophages to promote a supportive (immune-suppressed) tumor environment. These effects are mediated by GBM EV microRNAs such as microRNA-27a-3p and miR-21, although other EV components could certainly contribute towards this effect (Table 1, Figure 1).

Proteomics analysis has demonstrated a diverse set of proteins enriched in GBM EVs with likely bioactive effects [68]. Indeed, in 2018, Gabrusiewicz et al. also found that GBM stem-cell-derived exosomes (presumably small EVs) applied to monocytes induced a shift in monocyte-derived macrophage polarization state [49]. Specifically, through flow cytometry they found that macrophages exposed to these exosomes had reduced expression of M1 indicators (MHCII and CD80) and increased M2 indicators (CD163 and CD206). The authors found that the exosomes contained the transcription factor STAT3, a transcription factor heavily associated with M2 macrophage polarization [49,69]. This suggested that GBM-derived STAT3 is transferred to the macrophages inducing their phenotypic shift towards a tumor-supportive phenotype. Further, they demonstrated the exosomes used had a propensity to be taken up by macrophages relative to other cell types (Table 1, Figure 1) [49]. Additional work to understand how GBM-derived EVs target macrophage populations could unveil an advantageous therapeutic target.

Additional studies have also investigated the roles of GBM EVs on macrophage physiology. Yang et al. (2019) found that the GMB EV microRNA miR-214-5p was associated with poor clinical prognosis and targeted microglial *CXCR5* transcripts, and thus reduced protein expression (Table 1, Figure 1) [52]. CXCR5 has numerous roles in cancer biology, notably through chemoattractant interactions with its ligand CXCL13 [70]. Xu et al. (2021) found that exosomes (presumably small EVs) from hypoxic glioma cells also induce macrophages towards an M2 state through an induction of autophagy pathways (Table 1). These results were greatly subdued in EVs derived from normoxic glioma cells [53].

Collectively, it is clear that GBM EVs are taken up by macrophage populations and subsequently modulate cellular functions towards a tumor-supportive phenotype. While this is frequently addressed as an M2 macrophage or M2-like, it is likely far more complex, with numerous caveats. In 2015, de Vrij et al., for example, saw a macrophage shift towards the M2 phenotype after GBM EV exposure, yet observed increased IL-6 production. While IL-6 has been documented as a tumor-supportive cytokine in this context, it is typically associated with M1 macrophages [31,71]. This highlights the uniqueness of the GBM microglial/macrophage population to consider when developing macrophage-targeted therapies developed in other contexts.

## 5. Impact of Macrophage-Derived EVs on GBM Immunopathology

While GBM regulation of macrophage physiology is well documented, macrophage-derived EVs also influence tumor health. Under normal physiological conditions, macrophage EVs are a component of homeostatic immune function and cancer surveillance. For example, sEVs taken from healthy microglia displayed anti-cancer protective effects in a mouse model of GBM [54]. Specifically, microglia-derived sEVs modified glioma metabolism, thereby supporting the restoration of glutamate homeostasis [54], a key aspect of glioma pathophysiology [72]. Notably, these sEVs contained miR-124, an onco-suppressor highly downregulated in GBM patients’ tumors relative to healthy control normal brain tissue (Table 1, Figure 2) [54,73,74]. Unfortunately, following exposure to GBM EVs, the sEVs produced by macrophages shift towards a tumor-supportive phenotype, performing tumor-supportive functions [54]. Intriguingly, one mechanism through which macrophages are co-opted to support tumor growth is through their own sEV production. These macrophage sEVs then exert tumor-supportive functions through numerous distinct mechanisms detailed here.

Zhang et al. (2020), for example, demonstrated that tumor-associated macrophages produced sEVs containing a series of microRNAs (miR-27a-3p, miR-22-3p, and miR-221-3p) (Table 1, Figure 2) [55]. These microRNAs induced glioma stem cells to undergo proneural-to-mesenchymal transition through a CHD7-dependent mechanism. Proneural-to-mesenchymal transition during GBM progression is believed to be tumor-supportive, as it leads to increased resistance to various therapeutic modalities, including radiotherapy [55,75,76]. This suggests that GBM-derived EVs induce macrophages to produce EVs that in turn increase resistance to one of the few viable treatments clinically available to treat GBM. Interestingly, it is not yet known whether this process is instigated by mutagenic chance during GBM development or as a direct response to radiotherapy. Curiously, macrophage EVs from cervical cancer patients treated with radiation drove macrophages from M2-like to M1-like phenotypes. The impact of radiation on GBM-associated microglia and macrophages is complex, but usually considered detrimental to tumor treatment effects [77,78]; notably, this is often in the context of infiltrating macrophages following radiation [79]. The effect of microglial/macrophage-derived EVs in GBM therapeutic resistance seems understudied. Further insight into primary vs. recurrent GBM macrophage populations could begin to answer these questions.

Recently Zhao et al. (2022) implicated exosomes (likely small EVs) from M2-like tumor-associated macrophages in the phenotypes of glioblastoma stem cells. Specifically, they isolated glioblastoma stem cells and tumor-associated macrophages through fluorescence-activated cell sorting (FACS) of GMB tissue specimens. EVs isolated from the macrophage population were applied to the GBM stem cells where the EVs were found to maintain some stem cell properties of the glioblastoma stem cells (Table 1, Figure 2) [56]. Through an in vivo xenograft tumor model of GBM, they implicated the microRNA miR-27b-3p as the mediator for the macrophage EV-induced increase in the tumorigenicity of the glioblastoma stem cells. This miR-27b-3p-induced process was found to be mediated through the MLL4/PRDM1/IL-33 cell signaling axis [56].

Another key mechanism through which macrophage EVs support tumor growth is by supporting angiogenesis in the tumor microenvironment, allowing for the delivery of oxygen and metabolites to fuel further growth. One mechanism by which macrophage sEVs accomplish this is through the transfer of circular RNA circKIF18A from macrophage sEVs to brain microvessel endothelial cells, which enhances the activity of the FOXC2 transcription factor (Table 1, Figure 2) [57]. FOXC2 is a key regulator of angiogenesis [80]. Here, the authors attributed the pro-angiogenic effects of circKIF18A to the RNA-aiding nuclear entry of FOXC2, leading to the transcription factor upregulating the expression of ITGB3, CXCR4, and DLL4, and activating the PI3K/AKT signaling axis [57].

Azambuja et al. (2020) demonstrated that GBM EV-reprogramed macrophages produce EVs that promote glioma cell proliferation and migration in vitro [58]. Intriguingly, they attributed these effects, at least in part, to the expression of the anti-inflammatory enzyme Arginase-1 on the surface of these macrophage EVs (size not directly measured but likely sEVs) (Table 1, Figure 2). Arginase-1 is a potent anti-inflammatory mediator and is strongly associated with M2 macrophages [31,81]. Previous studies have identified Arginase-1 as an important mediator of the survival and progression of tumors [67,82]. Arginase-1 catalyzes the conversion of arginine to ornithine and urea. Local and systemic levels of these molecules have substantial impacts on inflammation, T-cell development and ability to recognize tumors, and general progression of some tumors [83]. For example, L-arginine is a key factor in the maturation of the T-cell receptor and is thus critical in the recognition of tumor antigens by T-cells. Arginase-1’s enzymatic function is to convert L-arginine into ornithine and urea; therefore, high Arginase-1 expression can directly deplete L-arginine and impair T-cell responses. This is in addition to Arginase-1’s roles in tumorigenesis and metastasis [84]. This novel observation of Arginase-1 on macrophage EVs, therefore, has significant implications in the tumor microenvironment [58].

Macrophage EVs are not inherently tumor-supportive; however, following exposure to GBM-derived EVs, macrophages adopt a polarization state that promotes tumor growth. While the numerous microRNAs and Arginase-1 described here are significant contributors to the growth-permissive effects of macrophage EVs, there are likely additional mechanisms contributing to the multifaceted macrophage response. With continued investigation, these pathways hold great potential in the development of future therapeutic targets.

## 6. Approaches to Target GBM/Macrophage EV Crosstalk

GMB/macrophage EV crosstalk has substantial impacts on GBM disease progression and therefore represents a potent therapeutic target. Given the complexities of GBM EV release, transfer to macrophages, and the subsequent production of EVs by macrophages, there are numerous aspects that could be therapeutically targeted, including EV neutralization, macrophage depletion, macrophage stimulants, exogenous EV therapies, and numerous other emerging approaches.

We have previously targeted the toxic effects of GMB sEVs on neurons using customized peptides targeting GBM sEVs, which we isolated with peptide phage display technology [85]. A similar approach could block the tumor-supportive effects of GBM sEVs on macrophage physiology and/or allow for additional anti-tumor cellular activity. This approach could be further enhanced through the use of multiple peptides in combination to maximize effects. Clearly, the delivery of such peptide blockers requires novel approaches, but one could envision application during surgery into the resection cavity [86], or perhaps even using healthy macrophage-derived EVs [87].

Macrophages perform predominantly detrimental functions in the GBM tumor environment as a result of macrophage manipulation by GBM sEVs [8,13,77]. While microglia and macrophages are capable of performing anti-tumor functions, in this case, it may be beneficial to remove them from the tumor microenvironment. One well-developed approach to deplete macrophages is through the use of clodronate-enclosed liposomes. These small artificial lipid vesicles are selectively taken by macrophages, minimizing off-target effects on other cells and tissues. When loaded with clodronate, the liposomes can deplete macrophages across numerous disease models, including a clinical trial in canines with soft-tissue sarcomas [88].

GBM-derived sEVs are a primary mediator suppressing macrophage anti-tumor activity [12]. One approach to rectify this could be macrophage-targeted therapies to shift macrophage polarization states towards anti-tumor activities. This could be performed through a systemic drug or a targeted approach, such as drug-enclosed liposomes. Stimulating macrophages with bacterial components (*Mycobacterium bovis* bacillus Calmette–Guérin), for example, have been found to induce increased anti-cancer macrophage activity [89]. Similarly, Banerjee et al. (2015) utilized *Mycobacterium indicus pranii* to repolarize macrophages towards an anti-tumor M1 phenotype [90]. Numerous other therapeutics are available with proven capabilities to modulate macrophage phenotype and anti-tumor activity [91]. With continued development, these therapeutics could be utilized to counter the detrimental effects of GBM-derived sEVs on macrophage physiology in the context of the GBM microenvironment.

Grimaldi et al. (2019) used an interesting approach to rescue microglia manipulated by GBM EVs. Microglia were exposed to the standard M1 stimulants lipopolysaccharide and interferon-gamma (LPS/IFN-γ), with microvesicles (EVs identified as 100–1000 nm, see [92]) isolated from the resulting culture. They then injected these treated EVs into the brains of tumor-bearing mice, resulting in a reduction in tumor size. Intriguingly, the M1 microglial-derived EVs resulted in a phenotypic shift of tumor macrophages towards an anti-cancer phenotype. Molecular analysis of the M1 microglia EV cargo identified a wide variety of inflammation-associated transcripts. While GBM patient microglia would be difficult to implement as a therapeutic approach, patient macrophages would be simple to culture from blood monocytes. Further work is needed to determine if this approach would be feasible or safe to develop as a treatment for GBM in humans. In a multi-layered approach, Wang et al. (2022) isolated sEVs from M1 macrophages that had been treated with AQ4N, a pro-drug that is activated in hypoxic conditions. AQ4N was sequestered in the released sEVs. These sEVs were further membrane-modified with two agents: a chemical excitation source, CPPO [hydrophobic bis(2,4,5-trichloro-6-carbopentoxyphenyl) oxalate]; and a photosensitizer, Ce6 (chlorin e6) [93]. Across multiple GBM models, the modified sEVs were able to induce M1 phenotypes in macrophages, leading to in vivo settings with intravenous injection of the modified sEVs. In addition to the anti-tumor effects of M1 macrophages, the increased H_2_O_2_ produced by M1 macrophages triggered a chemical reaction between CPPO and Ce6 creating cytotoxic reactive oxygen species within the tumor [93]. Further, the chemically driven induction of additional hypoxia released the activated anti-cancer agent AQ4. This unique approach, blending tumor microenvironment manipulation with localized chemical ablation of the tumor, holds great potential, but likely requires considerable development before reaching its clinical potential.

Given that GBM/macrophage EV crosstalk predominately supports pro-tumor functions, a blockade of these lines of communication may provide therapeutic benefits. Numerous compounds, chemicals, and peptides have been utilized to interfere with EV uptake [94]; however, the diverse nature of EV physiology presents a major challenge. The numerous subsets of EVs can each be taken up by several district cellular pathways, such as endocytosis, micropinocytosis, phagocytosis, membrane fusion, clathrin-dependent endocytosis, caveolin-mediated endocytosis, and lipid raft-mediated endocytosis [94]. To address this, further understanding of which EV populations are detrimental in GBM/macrophage EV crosstalk is needed. A targeted therapy, such as liposomal delivery to macrophages, may be able to bypass some of these challenges through combinatorial approaches targeting multiple uptake pathways.

Numerous publications have detailed the roles of sEV-derived microRNAs in pathological GBM/macrophage crosstalk [95,96], while another group has instead developed microRNA-loaded EVs as a potential GBM therapeutic in vitro. Specifically, Hong et al. (2021) loaded miRNA-124 into HEK293T-derived EVs (likely small EVs). In GBM cells (U373MG cells), these EVs reduced mRNA expression levels indicative of tumor progression, proliferation, migration, and epithelial–mesenchymal transition (STAT3, c-Myc, Mcl-1, ITG-β1, Vimentin, and Slug). In human microglia (SV40 microglial cell line), miRNA-124-loaded EVs (presumably small EVs) reduced expression of the M2 macrophage markers TGF-β and Arginase-1 and upregulated the M1 marker IL-6 [97]. It was noted that miR-504 appears downregulated in GBM and GBM stem cells compared to normal brain, particularly in those stem cells of the mesenchymal GBM subtype [98]. The miR targets the expression of Grb10, the latter acting as an oncogene. Forced over-expression of miR-504 in GBM stem cells reduced both their mesenchymal phenotype and their tumorigenicity. The miR could be passed from the GBM cells to microglia through EV secretion inducing a shift towards M1 polarization of the microglia, also likely reducing the pro-tumor capacities of the microglia (Bier et al., 2020). Similarly, Li et al. (2021) loaded long non-coding RNAs into exosomes (size not specified; protocol would suggest sEVs) as a means to demonstrate a mechanism in which GBM chemotherapy resistance develops through microglial complement production [99]. These studies highlight the adaptability of EVs as potential carriers of GBM therapeutics.

Extensive work across cancer disciplines has developed numerous approaches to target tumor-associated macrophages, many of which could be adapted to target GBM/macrophage EV crosstalk. Grégoire et al. (2020) performed a comprehensive review of clinical trials targeting macrophages as a therapeutic strategy to fight cancer [77]. These trials identified dozens of monoclonal antibodies and small molecule drug candidates aiming to block the recruitment of macrophages to the tumor, deplete macrophages from the tumor microenvironment, or reprogram macrophage polarization state towards anti-cancer phenotypes with Toll-like receptor (TLR) antagonists [77]. These studies emphasize the immense impact macrophages have on the tumor microenvironment and their impact as a therapeutic target. Similarly, Wang et al. (2022) reviewed recent clinical trials targeting macrophages, specifically for the treatment of GBM [17]. While these clinical trials were not performed with the explicit intent to target macrophage/GBM EV crosstalk, any approach that significantly depletes macrophages or blocks their recruitment would inevitably impact EV crosstalk. Similarly, any attempt to reprogram macrophages could also alter EV cargo that is impacted by macrophage activation state [100]. Intriguingly, many of these clinical trials aim to deplete macrophages or macrophage recruitment [77]; however, these approaches would not likely impact GBM-derived EVs. It is currently unknown whether the absence of macrophages could alter or enhance the GBM-derived EV impacts on the remaining cells in the tumor environment [101,102].

These diverse approaches to target GBM/macrophage EV crosstalk highlight the immense potential for the development of GBM therapeutics targeting these pathways; however, continued development is needed to push these approaches towards clinical applications.

## 7. Conclusions

GBM is a devastating disease in great need of new therapies to improve outcomes. GBM EV-mediated manipulation of macrophage immune defenses and reciprocal EV crosstalk from macrophages is a critical component impairing recovery; therefore, we have provided a brief description of these key mechanisms mediating direct GBM/macrophage/microglia EV crosstalk (Figure 1 and Figure 2, Table 1). To address the challenges produced through GBM EV crosstalk with macrophages and microglia, further insight is needed into the mechanisms through which GBM manipulates macrophage physiology to the tumor’s benefit. With this knowledge, we can better develop therapies to impair these detrimental processes and improve patient outcomes for this disease.

It is well established that GBM EVs are a primary mechanism through which macrophages are targeted [48,50,65]; however, much remains uncertain. EVs are incredibly diverse extracellular entities derived from several distinct populations and taken up through numerous cellular pathways [38]. Further understanding of which exact pathway(s) is/are mediating GBM control of macrophage populations is key for the development of future macrophage-directed therapeutics. Similarly, macrophages are a diverse heterogenous population of cells. Distinct roles and spatiotemporal contributions of both microglia and monocyte-derived macrophages likely have differential roles throughout the pathogenesis of GBM progression. Some of the above-mentioned studies do note the potential for recruitment or conversion of macrophages towards a tumor-attacking phenotype, suggesting that cells already within the tumor microenvironment may be therapeutically malleable. With continued understanding, therapeutics can be utilized to their greatest potential while minimizing unintended consequences.

Given the complexity of GBM/macrophage EV crosstalk, there are countless therapeutic targets and areas for proposed therapies. Unfortunately, most of these approaches are early in development and will require considerable time and investment before any potential clinical application. Given the dire need for new therapies for GBM patients, strong considerations should be made in the selection of therapeutic agents. Often, existing drugs can be repurposed, greatly improving the speed at which new therapies can be implemented. To date, numerous clinical trials across cancer disciplines have developed several distinct approaches to target tumor-associated macrophages [77]. Adapting successful approaches from other cancer disciplines into GBM therapies may allow for faster clinical implementation.

GBM/macrophage EV crosstalk is an important component of the GBM microenvironment that is relatively early in development. Given this, there remain many uncertainties and challenges to overcome. With continued development, however, these pathways hold great promise in the development of novel treatments for the GBM patient population for which there are currently so few currently available.

## Figures and Tables

**Figure 1 jcm-12-03430-f001:**
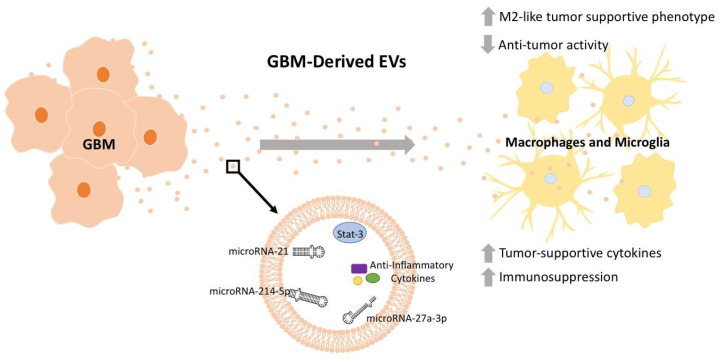
GBM-derived EV cargo manipulates macrophages and microglia physiology towards tumor-supportive functions. EVs are released from GBM tissue and have immunosuppressive effects on nearby macrophages and microglia, promoting their conversion towards an M2-like tumor-supportive phenotype. This effect on macrophages is mediated through EV cargo, including STAT3 [49], microRNA-214-5p [52], microRNA-27a-3p [50], and potential anti-inflammatory cytokines [49]. Depicted microRNA structures are not representative of actual structure.

**Figure 2 jcm-12-03430-f002:**
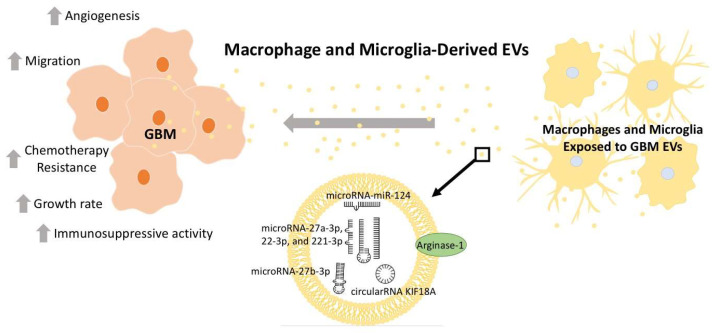
Macrophages and microglia exposed to GBM-derived EVs produce tumor-supportive EV cargo. These macrophages and microglia, modified by GBM EV-cargo, then release EVs that support tumor growth, migration, and chemotherapy resistance. This is mediated through EV cargo, including microRNA-124 [54,74], microRNA-27b-3p [56], microRNA-27a-3p [55], microRNA-22-3p and 221-3p [55], circularRNA-KIP18A [57], and Arginase-1 [58]. Collectively, this crosstalk suppresses the immune response to the tumor and supports further tumor growth. Depicted microRNA structures are not representative of actual structure.

**Table 1 jcm-12-03430-t001:** Key publications detailing direct GBM/macrophage/microglia EV crosstalk.

Author	Biological Mediator in EVs	Key Findings
de Vrij et al., 2015 [48]	Small RNA molecules	GBM-derived sEVs induced monocyte-derived macrophages towards an M2 pro-tumor phenotype. The macrophages produced increased levels of VEGF and IL-6.
Zhao et al., 2022 [50]	microRNA-27a-3p	Here, the authors demonstrated that microRNA-27a-3p inhibits enhancer of zeste homologue 1 (*EZH1*), thereby promoting M2 macrophage polarization. microRNA-27a-3p-treated macrophages had elevated levels of Arginase-1 (M2 phenotypic marker) and reduced iNOS (M1 phenotypic marker).
Abels et al., 2019 [51]	microRNA-21	The authors found that tumor-derived EVs deliver miR-21 to microglia. This resulted in a shift in numerous gene targets, notably the downregulation of *Pdcd4* and *Btg2*, causing increased microglial proliferation.
Gabrusiewicz et al., 2018 [49]	Stat3	GBM stem-cell-derived sEVs applied to monocytes induced a shift in monocyte-derived macrophage polarization state towards a tumor-supportive M2 phenotype. This was evidenced through flow cytometry analysis, indicating reduced expression of M1 indicators (MHCII and CD80) and increased M2 indicators (CD163 and CD206). The sEVs contained the transcription factor STAT3, which is associated with M2 polarization.
Yang et al., 2019 [52]	microRNA-214-5p	GMB EV microRNA miR-214-5p was associated with poor clinical prognosis and targeted microglial *CXCR5* transcripts, and thus reduced CXCR5 protein expression.
Xu et al., 2021 [53]	Stat3, IL-6, and microRNA-155-3p	GBM-derived sEVs were isolated under hypoxic conditions, common in the GBM microenvironment. These EVs induce macrophages towards an M2 state through an induction of autophagy pathways. These results were greatly subdued in EVs derived from normoxic glioma cells.
Serpe et al., 2021 [54]	microRNA-124	Cultured microglia were stimulated with LPS and IFNγ (M1 activation). sEVs isolated from these microglia were applied intracranially via a cannula infusion into tumor-bearing mice. The application of these microglia-derived EVs resulted in prolonged survival and reduced tumor mass. These effects were attributed to microRNA-124-induced modulation of tumor metabolism.
Zhang et al., 2020 [55]	microRNA-27a-3p, microRNA-22-3p, and microRNA-221-3p	The authors found that tumor-associated macrophages produce sEVs that trigger a proneural-to-mesenchymal transition in glioma stem cells. This is believed to be tumor-supportive as it leads to increased resistance to various therapeutic modalities, including radiotherapy. This was attributed to EV cargo, particularly microRNA-27a-3p, microRNA-22-3p, and microRNA-221-3p.
Zhao, G et al., 2022 [56]	microRNA-27b-3p	The authors isolated glioblastoma stem cells and tumor-associated macrophages from GMB tissue specimens. Macrophage sEVs were applied to the GBM stem cells where they maintained some of the stem cell properties of the GBM stem cells. They implicated microRNA-27b-3p for the increase in the tumorigenicity of the GBM stem cells. This process was found to be mediated through the MLL4/PRDM1/IL-33 cell signaling axis.
Jiang e al. 2022 [57]	Circular RNA circKIF18A	Here, a human microglial cell line was treated with GBM-conditioned media to induce an M2 phenotype. sEVs from these treated microglia then transferred circular RNA circKIF18A from microglia. circKIF18A enhances the FOXC2 transcription factor activity, leading to increased angiogenesis in the tumor microenvironment.
Azambuja et al., 2020 [58]	Arginase-1	Using an in vitro model of GBM/macrophage crosstalk, the authors found that macrophage EVs promote glioma cell proliferation and migration in vitro. Using chemical inhibitors, they identified Arginase-1 on the surface of these EVs as a primary mediator. Arginase-1 is strongly associated with M2 macrophages.

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
