# Peer review of "Immunopathology of Extracellular Vesicles in Macrophage and Glioma Cross-Talk"

_jcm, 2023, doi:10.3390/jcm12103430_

Round 1

Reviewer 1 Report (Previous Reviewer 1)

I have reviewed the manuscript entitled "Cross-talk Between Macrophage/Microglia and Gliomas: Roles of Extracellular Vesicles in Glioma Immunopathology" submitted to the Journal of Clinical Medicine. Overall, I find the idea behind the review article to be interesting and relevant to those working on innate immunity. However, I do have some comments and suggestions that I believe would improve the manuscript.

1- Firstly, regarding the title of the manuscript, I believe it could benefit from some rephrasing to make it more concise and easier to understand. While the current title conveys the main topic of the manuscript, it may be helpful to include more specific information about the focus and scope of the review. This could help readers better understand the contents of the article and its relevance to their own research.

2- Furthermore, before discussing the polarization of macrophages in glioblastoma (GBM), I suggest that more information about the role of macrophages in innate immunity and malignancies should be included. This would provide a better context for understanding the significance of macrophage polarization in GBM and its potential implications for cancer immunotherapy.

3- In order to provide a more comprehensive review of the topic, I suggest that the molecular mechanisms involved in macrophage responses in favor or against GBM should be listed thoroughly. This would help readers better understand the complex interactions between macrophages and GBM and the potential targets for therapeutic intervention.

4- Finally, To more effectively illustrate the interaction between extracellular vesicles (EVs) and macrophages, it is imperative to employ figures of superior quality. As such, it is recommended that utmost care be taken to ensure that the images utilized are clear, detailed, and accurately represent the underlying biological processes. By doing so, a more cogent and convincing visual representation of EV-macrophage interaction can be achieved, thereby enhancing the overall quality and impact of the research findings.

Author Response

Reviewer 2 Report (Previous Reviewer 2)

Kopper et al have provided a revision of the critical aspect highlighted in the previous round of revision. However, there are still some concerns that have arisen after the changes performed by the authors.

References to table1 should probably go in section 4, and as one example of the multiple ones:

In 2015 de Vrij et al. applied GBM-derived EVs (presumably small EVs) onto monocyte-derived macrophages and observed several changes [42]. (Table 1).

The locations of this Table 1 should be close to the place where it has been cited for the first time.

The sentence at the end of section 3 doesn't make much sense, however, if the authors want to keep it, they should place the table there. Accordingly, the sentence doesn’t need to refer to the location of the Table anymore since it would already be there. Table 1, at the conclusion of this review, provides a brief summary of the key mechanisms through which GBM tissue and macrophages exchange EV crosstalk Table 1 provides a brief summary of the key mechanisms through which GBM tissue and macrophages exchange EV crosstalk.

Figure 1 should be cited whenever the information contained there is mentioned. As one example of the multiple ones:

Inhibition of EZH1 has been previously demonstrated to promote M2 macrophage polarization [52]. Indeed, the authors observed that microRNA-27a-3p treated macrophages had elevated levels of Arginase-1 (M2 phenotypic marker) and reduced iNOS (M1 phenotypic marker). (Fig. 1).

For the same, the location of this Figure should be close to the place where it has been cited for the first time. Figure 2 must be also cited in the proper places and its location should be changed.

Given the number of data that has to be cited in this sentence, makes more sense to indicate it in brackets:

GBM is a devastating disease in great need of new therapies to improve outcomes. GBM EV-mediated manipulation of macrophage immune defenses and reciprocal EV crosstalk from macrophages is a critical component impairing recovery as detailed in Figure 1 and Figure 2. Table 1 provides a brief description of these key mechanisms mediating direct GBM/macrophage/microglia EV crosstalk.--> GBM is a devastating disease in great need of new therapies to improve outcomes. GBM EV-mediated manipulation of macrophage immune defenses and reciprocal EV crosstalk from macrophages is a critical component impairing recovery, therefore we have provided a brief description of these key mechanisms mediating direct GBM/macrophage/microglia EV crosstalk (Fig 1-2, Table 1).

Author Response

This manuscript is a resubmission of an earlier submission. The following is a list of the peer review reports and author responses from that submission.

Round 1

Reviewer 1 Report

Kopper et al. reviewed the mechanisms through which GBM-derived EVs impair macrophage function, how subsequent macrophage-derived EVs support tumor growth, and the current therapeutic approaches to target GBM/macrophage EV crosstalk.

Although the current review is a significant contribution to the field, minor issues should be addressed.

1- Regarding the macrophage/microglial polarization in GBM, a figure describing the interaction of microglial cells during GBM and what regulators affect the polarization status should be added. Moreover, more original articles should be discussed in this section, especially those aimed at evaluating the polarization status of macrophages. The underlying mechanisms and molecular regulators and transcription factors could be introduced here.

2- There is enough information regarding the impact of GBM-derived EVs on macrophage function. However, a figure is missing here, which could better describe the mechanism. This figure could be combined with the impact of macrophage-derived EVs on GBM immunopathology.

3- Regarding the approaches to target GBM/macrophage EV crosstalk, please prepare a table showing the ongoing treatments and clinical trials. 

Reviewer 2 Report

In this article Kopper and colleagues have reviewed the mechanisms through which GBM-derived EVs impair macrophage function, how subsequent macrophage-derived EVs support tumor growth, and the current therapeutic approaches to target GBM/macrophage EV crosstalk. Even if I think the topic is really interesting and deserve to be reviewed, the authors should invest a bit more time in building an article that really covers all the aspects. As main drawbacks, the article is really short, lack of images that help to understand it, and it is not well referenced. Therefore, I recommend a resubmission of the article.

Some of the mentioned aspect and others:

- In the field, there is a clear shift on the terminology from exosomes towards small EVs. Maybe the authors should comment on that to be able to freely unify the terms and use small EVs even if the original publication uses the word exosomes. Right know the use of both terms it is pretty messy in the review.

 - The authors have to properly indicate the subject of the action. To illustrate that:

“Additional work to understand how GBMs target macrophage populations could unveil an advantageous therapeutic target”à Additional work to understand how GBMs derived small EVs target macrophage populations could unveil an advantageous therapeutic target”. Here, just writing GBMs doesn´t mean anything.

- Some figures to illustrate the content of the review and make it clearer. Tables for example summarizing the EVs content (i.e. miRNAs ), short description of the experiment, the effect and indicating the references would be of help to the reader.

- 65 references doesn´t seem to be a great number for a review article. When reading the manuscript, it is clear that the facts are nor properly referenced. As an example, the introduction has only 6 references. More that half of it has absolutely no references with sentences such as “In an increasing number of instances, this is induced through extracellular vesicle (EVs) intercellular communication” with no citations.

Round 2

Reviewer 2 Report

Kopper et al have tried to answer the suggested revisions. The resulting manuscript is difficult to read since they didn’t attach a clean version either. The authors tried to do some shortcuts and that is clearly shown in the new version.

Major concern:

- It would have been nice to have separately:

1- Figure showing the interaction of microglial cells during GBM and with the known regulators. Reflecting the information on 2. Macrophage/microglial polarization in GBM pathology

2- Figure with mechanisms through which GBM EVs manipulate macrophage polarization (which is different to the previous one). Summarizing the data in 4. Impact of GBM-derived EVs on macrophage function.

3-Table with a short description of the EVs content (i.e. miRNAs ), and the performed experiment as previously suggested.

As it is right now it is very messy and it is difficult to extract the information from that only figure.

- After explaining the dilemma about terminology, it is not clear that every time they should discuss the application of the guidelines ( i.e. “exosomes (sEVs by MISEV2018 guidelines)” ) instead of applying the terminology at the beginning.

- Since the authors have updated the core of the review with reviews about clinical trials, this should be also updated in the conclusion part where the clinical trials are mentioned but not discussed as it has not been changed from the previous version.